# The Microstructural Evolution and Corrosion Behavior of Zn-Mg Alloys and Hybrids Processed Using High-Pressure Torsion

**DOI:** 10.3390/ma17010270

**Published:** 2024-01-04

**Authors:** Ayoub Tanji, Hendra Hermawan, Carl J. Boehlert

**Affiliations:** 1Department of Mining, Metallurgical and Materials Engineering, Laval University, Quebec City, QC G1V 0A6, Canada; ayoub.tanji.1@ulaval.ca (A.T.); hendra.hermawan@gmn.ulaval.ca (H.H.); 2Department of Chemical Engineering and Materials Science, Michigan State University, East Lansing, MI 48824, USA

**Keywords:** severe plastic deformation, microstructure, corrosion

## Abstract

Zinc (Zn) alloys, particularly those incorporating magnesium (Mg), have been explored as potential bioabsorbable metals. However, there is a continued need to enhance the corrosion characteristics of Zn-Mg alloys to fulfill the requirements for biodegradable implants. This work involves a corrosion behavior comparison between severe-plastic-deformation (SPD) processed cast Zn-Mg alloys and their hybrid counterparts, having equivalent nominal compositions. The SPD processing technique used was high-pressure torsion (HPT), and the corrosion behavior was studied as a function of the number of turns (1, 5, 15) for the Zn-3Mg (wt.%) alloy and hybrid and as a function of composition (Mg contents of 3, 10, 30 wt.%) for the hybrid after 15 turns. The results indicated that HPT led to multimodal grain size distributions of ultrafine Mg-rich grains containing MgZn_2_ and Mg_2_Zn_11_ nanoscale intermetallics in a matrix of coarser dislocation-free Zn-rich grains. A greater number of turns resulted in greater corrosion resistance because of the formation of the intermetallic phases. The HPT hybrid was more corrosion resistant than its alloy counterpart because it tended to form the intermetallics more readily than the alloy due to the inhomogeneous conditions of the materials before the HPT processing as well as the non-equilibrium conditions imposed during the HPT processing. The HPT hybrids with greater Mg contents were less corrosion resistant because the addition of Mg led to less noble behavior.

## 1. Introduction

Over the last two decades, the field of biomedical implants has garnered considerable attention due to its potential to enhance human life (both quality and longevity) by replacing damaged hard and soft tissues [1]. Metallic implants have been widely used as orthopedic and vascular devices due to their high strength and fracture toughness compared with polymers and ceramic materials. Traditional metallic biomaterials with high corrosion resistance, such as stainless steel, are typically considered for permanent implant applications [2]. However, in many cases, the function of the implant is temporary and, therefore, not needed after full recovery. Moreover, the permanent presence of the implant in the body upon healing can develop multiple complications. For instance, the release of metal ions from the implant device over time can lead to the accumulation of trace metals in the surrounding tissues, which can in turn trigger a chronic inflammatory response. A second surgery is sometimes necessary for implant extraction and/or replacement, resulting in additional injury, expense, and discomfort, as well as an extra risk of infection [2,3]. In addition, implant failure can occur in bone fixation devices due to the stress-shielding effect, which refers to the fact that implants carry most of the mechanical load, impeding load transfer to the bone tissue, which is necessary to stimulate new bone formation [4,5]. This occurs when the stiffness of the implant material is substantially higher than the bones to be replaced, which is often the case for the metals typically used for permanent applications.

Bioabsorbable metals with relatively low stiffness values offer the opportunity to design temporary implant devices able to overcome the above-mentioned limitations. It is desirable for absorbable metal implants to maintain mechanical integrity during tissue healing, and then, corrode gradually and safely over time, while progressively transferring the mechanical load to the healed tissue [6,7]. Therefore, absorbable metal implants must maintain strength and corrosion resistance in a physiological environment while maintaining their biocompatibility. Iron (Fe) and magnesium (Mg), both pure and alloyed, have been extensively studied as potential absorbable metals for biomedical applications [8,9]. However, the large number of research studies with these material systems has identified limitations in terms of their suitability for clinical applications. On the one hand, Fe and Fe-based alloys generally exhibit corrosion rates below the clinical needs, which unnecessarily prolongs the exposure time of the organism to the implant after the healing and recovery period [10]. In addition, the corrosion products of Fe tend to be stable in the physiological environment, resulting in long-term retention, causing metabolic complications, rather than promoting cells to integrate around the implant [11]. Mg and Mg alloys typically exhibit inadequate mechanical strength, as well as high corrosion rates in physiological environments. The latter issue is particularly detrimental, as it leads to the formation of detrimental hydrogen gas pockets near the implant [12]. Therefore, recognizing the substantial research efforts and knowledge gained on Fe and Mg alloys for implant applications, there is still the need for metallic materials able to satisfy the demanding set of requirements of absorbable implant applications.

Zinc (Zn) has emerged as an alternative metal for absorbable medical implants [13]. The following are some aspects that make Zn and Zn-based alloys especially attractive for use in temporary clinical applications. Similar to Fe and Mg, Zn is an essential element in the human body, critical for a significant number of metabolic activities and cellular functionalities, including cell growth, proliferation, and differentiation. Thus, the Zn ions released from the implant during degradation are expected to integrate into the normal metabolic activity of the host without causing detrimental side effects [2,14]. The standard electrode potential of Zn is −0.76 V, which is between that of Mg (−2.37 V) and Fe (−0.44 V). This indicates that Zn exhibits moderate corrosion rates (faster than the slowly degrading Fe and its alloys, but slower than the rapidly degrading Mg), and this is caused by the corrosion products forming passive layers of moderate stability [13,15]. In general, Zn and Zn alloys can be more easily processed than Mg and Fe alloys [13,16].

It has been shown that Zn exhibits near ideal biocorrosion and biocompatibility for use as biodegradable endovascular stents [17,18]. This discovery holds promise for fulfilling the dream of the biomedical devices industry to replace permanent stents with stents that could perform their function in the first few months and then dissolve in the host’s body, eliminating the harmful long-term effects experienced with the current permanent stents. However, the primary drawback of pure Zn as an absorbable stent material is its lack of mechanical strength. The room temperature (RT) ultimate tensile strength (UTS) of pure, drawn Zn wires is ~120 MPa, whereas the ideal UTS for an absorbable stent is approximately 300 MPa [17]. Thus, the tensile strength of Zn is not ideal for stents. Alloying can be used to increase the UTS while maintaining high elongation-to-failure (ε_f_) values and corrosion resistance, as many commercial Zn alloys possess UTS values of approximately 300 MPa and exhibit > 20% ε_f_. Pure Zn has a lower RT strength compared to Mg, but once alloyed, such as in Zn-(1–3 wt.%)Mg, its strength can be superior to some Mg alloys [19]. Moreover, alloying Zn with Mg (<4 wt.%) enhances the corrosion resistance [20]. Grain refinement can also enhance properties, including allowing more mechanical isotropy and increasing the yield strength (YS) and UTS [21]. Grain refinement to improve the ε_f_, UTS, and corrosion uniformity of Zn, without sacrificing the corrosion rate, could very well pave the way to a fully biodegradable Zn-based stent. The microstructures of extruded metals and alloys, such as those based on Zn, are often refined by mechanical means [22], such as through severe plastic deformation (SPD) techniques including equal-channel angular pressing (ECAP) [23,24,25] and high-pressure torsion (HPT) [26]. In this study, HPT, which results in significant grain refinement [27,28], was used to stabilize a highly refined microstructure exhibiting high hardness. HPT has been performed on pure metals [29,30,31,32,33,34,35,36], alloys [37,38,39,40], layers of dissimilar metal disks [41,42,43,44,45,46,47,48,49,50,51], and powders [52].

A Zn-based biomaterial with a highly refined microstructure, such as that resulting from HPT, will undoubtedly change the corrosion behavior of the implant and its resultant biocompatibility. It is difficult to speculate on the expected effects on corrosion due to the discordant nature of previous studies on Zn [53]. Some investigators have observed enhanced corrosion resistance in nanostructured Zn [54], while others have reported that the higher grain boundary density provides more sites for corrosion initiation, and thus, enhances degradation [55]. It is conceivable that both of these effects could compete in the complicated physiological environment, and it is not trivial to predict which effect will dominate. It can be supposed, at least, that a high density of grain boundaries will promote uniformity of corrosion or passivation, a desirable trait in biodegradable materials. The complex set of interactions between the aqueous environment and the fine-grained material will require careful appraisal via corrosion experiments. This work compares the corrosion behavior of different HPT-processed materials, both after HPT and after HPT followed by a post-deformation annealing (PDA) treatment. The effect of Mg content (3, 10, and 30 wt.%) on the corrosion behavior was studied using identical processing conditions (i.e., 15 turns) for the HPT hybrids. The HPT-processed alloys were called “HPT alloy” and its HPT-processed hybrids were called “HPT hybrid”.

## 2. Materials and Methods

### 2.1. Processing

This study used 10 mm diameter cast bars of the following compositions: Zn-3Mg (wt.%), Zn-10Mg (wt.%), Zn-30Mg (wt.%), commercially pure Zn (99.97% purity), and commercially pure Mg (99.90% purity). Henceforth, all compositions are provided in wt.%. The as-cast Zn-3Mg alloy underwent a homogenization heat treatment in a vacuum tube furnace at 360 ± 2 °C for 15 h followed by water quenching (WQ) prior to HPT processing. For the HPT alloy samples, 1.5 mm thickness disks were sliced from the respective Zn-Mg bar castings prior to HPT processing. For the HPT hybrid samples, Zn and Mg disks were sliced and stacked on top of each other in the order of Zn/Mg/Zn so that the targeted bulk composition of the resulting HPT hybrid sample was Zn-3Mg, Zn-10Mg, or Zn-30Mg. The hybrid samples exhibited a heterostructure across the disk diameter [41,42,43,44,45,46,47,48,49,50,51], thus, the composition is estimated as an average for the entire disk volume.

HPT processing was performed at RT under an applied pressure of 6 GPa, a constant rotational speed of 1 rpm, and for either 1, 5, 15, or 30 turns [56,57]. The temperature rise (<50 °C) during HPT processing was not considered high enough to significantly affect the microstructural evolution, in agreement with [58]. After HPT processing, the HPT 30 turns alloy was then subjected to PDA at 200 °C for 1 h in a vacuum tube furnace followed by air cooling. HPT-processed disks were cut along their diameter and the resulting cross-sections were mechanically polished using a 0.04 μm particle size.

### 2.2. Microstructural Characterization

Scanning electron microscopy (SEM), electron backscattered diffraction (EBSD), and energy-dispersive spectroscopy (EDS) were performed using a Tescan Mira3 FEG-SEM (Tescan Group, Kohoutovice, Czech Republic). Secondary electron (SE) and backscattered electron (BSE) images were acquired. X-ray diffraction (XRD) analysis was performed at the mid-thickness of the shear-radial plane using a high-resolution Rigaku Ultima IV (Rigaku Co., Tokyo, Japan) to investigate the phases present in the microstructures. The XRD data were collected with an accelerating voltage of 44 kV and a beam intensity of 44 mA, within the range 20° < 2θ < 90° using Cu-Kα radiation in a Bragg-Brentano configuration. All XRD scans were collected with a step size of 0.01° and a scanning speed of 3°/min. The following ICSD material files were used for the analysis: Zn (421014), Mg (77908), Mg_2_Zn_11_ (104898), and MgZn_2_ (104897). For the Zn-10Mg HPT hybrids, after identification of the phases present, their amounts were quantified by means of Rietveld analysis by following the Rigaku PDXL User Manual version 2.6. The background signal was subtracted, and peaks in the XRD spectra were identified by comparison with ICSD files. After confirming the phases present in the material, the theoretical diffraction pattern was simulated and displayed together with the experimental pattern. The residual intensity plot indicated the relative differences in peak heights between the theoretical and experimental XRD patterns. After a series of iterations that adjusted the relative intensity ratios of each phase as well as the scale factor, the residual intensity was reduced. Once both theoretical and experimental patterns were brought to be practically coincident, the refinement processed was completed, and the quantitative information of each phase, in wt.%, was provided.

### 2.3. Corrosion Behavior Evaluation

The disk specimens were cut vertically along their diameter to give two semicircular disks, then the cross-section of each disk was mounted and mechanically ground. Subsequently, they were polished using a polycrystalline diamond paste of 6, 3, and 1 μm particle size, and a colloidal SiO_2_ suspension with a particle size of 0.05 μm. A model 760E potentiostat (CH Instruments, Austin, TX, USA) was employed in a three-electrode configuration, involving a test specimen, a saturated calomel electrode (SCE), and a graphite rod as the working, reference, and counter electrodes, respectively. All potential measurements were referenced to SCE. Three distinct methods were employed: open circuit potential (OCP), potentiodynamic polarization (PDP), and electrochemical impedance spectroscopy (EIS). All experiments were performed in 100 mL Hanks’ solution maintained at 37 ± 1 °C with a pH of 7.2. The solution composition (in mg/L) was the following: 400 KCl, 60 KH_2_PO_4_, 8000 NaCl, 350 NaHCO_3_, 48 Na_2_HPO_4_, and 1000 D-glucose. A stable OCP was achieved after a stabilization time of 1 h, marking the commencement of each experiment. The EIS was performed in a frequency range of 10^5^ to 10^−2^ Hz, centered around the OCP, with a peak-to-peak amplitude perturbation of 0.01 V, while during the PDP, a scan rate of 1 mV/s was applied. To ensure reliability, each experiment was performed in triplicate. Data acquisition and analysis were carried out using the software of CH Instruments and EC-Lab version 10.4 (BioLogic, Seyssinet-Pariset, France), respectively.

## 3. Results

### 3.1. Microstructure of the Zn-3Mg Alloy after the Homogenization Heat Treatment

Figure 1 shows the microstructure of the heat-treated Zn-3Mg alloy. The chemical composition of the light and dark phases was investigated using a series of EDS measurements. The average chemical composition of the darker phase, as measured by EDS, was 5.8 ± 0.1 wt.% Mg and 94.2 ± 0.1 wt.% Zn [58]. This composition is consistent with that of the Mg_2_Zn_11_ phase. The average chemical composition of the brighter phase was close to 100% Zn [59].

### 3.2. Microstructural of the Alloy and Hybrid as a Function of the Number of Turns and the Mg Content

Figure 2a shows a series of low-magnification SE-SEM micrographs taken of the cross-sections of the Zn-3Mg hybrids after HPT for 1, 5, 15, and 30 turns. The sample after 30 turns that was subjected to a PDA at 200 °C for 1 h (termed ‘30 turns HPT + PDA’) is shown at the bottom of Figure 2a. For the one-turn sample, the intermediate Mg layer remained evident. The Mg layer was no longer continuous after five turns. The Mg-rich phase was fragmented into thin regions, especially for the regions where larger plastic deformation occurred (i.e., closer to the edges). The central region of the disks exhibited finer Mg-rich phases for the 15-turn and 30-turn samples. The microstructure at the disk edges became more homogeneous with an increasing number of turns. No macroscopic evidence of Mg-rich phases was evident after 30 turns. For the 30-turn HPT + PDA sample, a complete mixture of the phases was observed. Figure 2b presents low-magnification SE-SEM photomicrographs taken of the cross-sections of the Zn-3Mg alloys as a function of the number of turns. In comparison to the HPT hybrids, the HPT alloys exhibited a more homogeneous distribution of the Zn-rich and Mg-rich phases throughout the cross-sections. In both the hybrids and alloys, significant grain refinement was caused by HPT as the grain size for both phases was between 200 and 400 nm. For example, after HPT, the grain size of the Zn-3Mg alloy (combining both phases) after 30 turns was 213 ± 59 nm [59].

After PDA, the microstructure showed no macroscopic evidence of different phases. It should be noted that the large Mg-rich phases present in the central region of the sample after 30 turns were no longer distinguishable after PDA, Figure 2a. However, the higher-magnification SE-SEM photomicrographs in Figure 3a–c depicted grain sizes on the order of 700 nm or larger for both the Zn-rich and Mg-rich phases. It is noted that this was approximately three-times larger than the grain size before the PDA treatment [59]. It is notable that the Mg-rich regions, highlighted with blue-dashed circles in Figure 3c, were cluttered with light nanoscaled precipitates. High-magnification BSE-SEM photomicrographs of the Zn-3Mg alloy microstructures are provided in Figure 3d–f. The Zn-rich phase volume percent was approximately 68%. The remainder of the microstructure (~32 volume percent) contained the Mg_2_Zn_11_ phase regions.

Figure 4a,b show SE-SEM photomicrographs of the Zn-10Mg HPT hybrid after 15 turns. It is clear that a higher volume fraction of the Mg-rich phases was present compared to the Zn-3Mg HPT disks. A cross-sectional SE-SEM photomicrograph of the Zn-30Mg HPT hybrid after 15 turns is presented in Figure 4c.

Figure 5 depicts a plot of intensity versus two-theta from the XRD analysis of the Zn-10Mg hybrid after 1 turn and 15 turns, and the associated Rietveld analysis of the phases present is depicted in Table 1.

The disks were well-bonded in the middle of that sample. The Mg disk fragmented into smaller phases at r > 2 mm, but not in the central region of that sample. Both Zn-rich and Mg-rich phases were still evident in relatively large regions, leading to notably more macroscopic heterogeneity in the Zn-30Mg HPT hybrids compared to the Zn-10Mg and Zn-3Mg HPT hybrids. Thus, larger Mg contents lead to more microstructural inhomogeneity.

### 3.3. Effect of Mg Content on the Corrosion Behavior of HPT Hybrid Samples

The electrochemical behavior of hybrid specimens after 15 turns was notably influenced by the Mg content. This influence becomes evident through the results presented in Figure 6 and Table 2. In terms of potential–time curves, both the Zn-3Mg and Zn-10Mg specimens exhibited a consistent increase in potential over time. This suggests the formation of a passive film on the surfaces of these specimens. In contrast, the Zn-30Mg specimen exhibited a declining potential over time, indicating a surface that tends to be more active than the other specimens. The formation of passive films on the surfaces of the Zn-3Mg and Zn-10Mg specimens was apparent from their potentiodynamic polarization results, where a passive domain was present in these specimens. This formation is characterized by a strong disturbance as presented by the OCP curves, indicating that this passive film is not stable. A relationship between Mg content and corrosion-related factors was evident. As the Mg content increased, the corrosion potential decreased, signifying an augmented susceptibility to corrosion. This trend was further confirmed by the increase in corrosion current density with a higher Mg content, indicating a higher corrosion rate. Examining the impedance characteristics, it is evident that as the Mg content increased, there is a noticeable reduction in the resistive–capacitive attributes observed in the Nyquist plot, Figure 6c. Additionally, the charge transfer resistance decreased with increasing Mg content, indicating high electrochemical activity. The phase angle behavior assumes significance, notably in its approach towards 0° at intermediate- and low-frequency regions. This pattern is often associated with the presence of a weak passive film and an active surface. Such behavior aligns with the non-linear variation in the impedance modulus.

A proposed equivalent circuit model, shown in Figure 6e, comprises distinct components: resistances for the solution, surface film, and charge transfer (R_s_, R_f_, R_ct_, respectively), and introduces constant phase elements to characterize the surface film (CPE_f_) and double layer (CPE_dl_). Notably, the choice of employing a constant phase element, rather than a capacitor, to encapsulate the system’s behavior stems from its ability to represent non-ideal characteristics (such as variations in conductivity due to thickness fluctuations). The parameter “n” associated with CPE_f_ signifies the non-uniform current distribution arising from surface heterogeneity, encompassing factors like roughness and defects.

### 3.4. Effect of Number of Turns on the Corrosion Behavior

The impact of the number of turns on the electrochemical characteristics of alloy specimens, while keeping the reference content of Zn-3Mg constant, appeared to be negligible, as demonstrated in both Figure 7 and Table 3. Both the potential and corrosion current density exhibited no substantial variation with the changes in the number of turns. Likewise, the passive domain observed in the potentiodynamic polarization results remained small and exhibited minimal change as the number of turns increased. This pattern extends to the charge transfer resistance and passive film properties as well, where the number of turns did not seem to exert a noteworthy influence.

The influence of the number of turns on the electrochemical traits of the hybrid specimens, while maintaining a consistent reference content of Zn-3Mg, shows noteworthy effects, as presented in both Figure 8 and Table 4. Alterations in the number of turns within the hybrid process exerted a substantial impact on corrosion-related characteristics. Specifically, an increase in the number of turns corresponded to a rise in the open circuit potential as well as an escalation in metastable pitting behavior. Moreover, elevating the number of turns was associated with a heightened corrosion potential, a reduction in corrosive activity, and an increase in the charge transfer resistance. Additionally, the number of turns contributed to the modification of the resistive–capacitive properties, evident through the enlargement of the semi-circles in the Nyquist plot in Figure 8c.

### 3.5. Effect of Synthesis Approach on the Corrosion Behavior of Zn-3Mg

The effect of the material process on the electrochemical characteristics, while maintaining a consistent reference content of Zn-3Mg, yields distinct outcomes, as shown in both Figure 9 and Table 5. Notably, the alloying (casting) process contributed to an increase in the corrosion resistance of the material when contrasted with the hybrid process. In terms of potentiodynamic polarization, alloy specimens display a notably smaller passive domain compared to that formed by the hybrid process. Unlike the hybrid process, the potentiodynamic polarization curves for the casting did not exhibit metastable pitting. However, the hybrid process results in elevated corrosion current densities. Furthermore, the alloying process leads to enlarged semi-circles in the Nyquist plot, Figure 9c, indicative of heightened resistive–capacitive properties.

## 4. Discussion

### 4.1. Effect of Homogenization Treatment on the Zn-3Mg Alloy

The homogenization treatment provided a more homogeneous microstructure than the Zn-3Mg casting [59]. The composition of the darker phases in the microstructure portrayed in Figure 1 was in agreement with that of the Mg_2_Zn_11_ intermetallic phase: 93.7Zn-6.3Mg.

### 4.2. Microstructure of the Zn-3Mg HPT Alloys and Hybrids and Effect of Subsequent PDA on the Alloy (All after 30 Turns)

The microstructure of the Zn-3Mg alloy after 30 turns of HPT processing consisted of a matrix of Zn-rich and Mg_2_Zn_11_ grains with sizes ranging between ~100 and 300 nm. HRTEM images confirmed the presence of nanocrystalline domains [59]. Upon PDA, the average grain size of the alloy increased to ~700 nm. The Mg_2_Zn_11_ nanoscale intermetallics, which were either clustered between neighboring Zn grains, or located along the grain boundaries after HPT, diffused during PDA and coalesced into larger grains. The volume percent of Zn and Mg_2_Zn_11_ was estimated as 68% and 32%, respectively, which are close to the theoretical equilibrium composition of 57Zn-43Mg_2_Zn_11_ for Zn-3Mg.

### 4.3. Processing and Microstructure Effects on the Corrosion Behavior

The examination of various processing and microstructure variables and comparison of their varying degrees of influence on the corrosion behavior reveals that the hybrid specimens with 30Mg and 15 turns exhibit the lowest corrosion potential and the highest corrosion current density (−1.51 ± 4 × 10^−3^ V and 358.3 ± 30.2 μA·cm^−2^, respectively; Table 2). Conversely, the highest corrosion potential and the lowest corrosion current density were obtained by the hybrid specimens with 3Mg and 15 turns (−1.19 ± 48 × 10^−3^ V and 13.4 ± 5.5 μA·cm^−2^, respectively; Table 4). However, an additional polarization test on the annealed 3Mg alloy demonstrated an even lower corrosion current density of 8.4 ± 0.7 μA·cm^−2^. In terms of surface characteristics, the alloy specimens show n_f_ values close to 1 and lower Q_f_ values, indicating a higher surface film compactness and reduced roughness when compared to the hybrid specimens (Table 5). Upon considering the overall combined effect of various parameters, a clear trend emerges. The combination of a hybrid process, along with a low number of turns (1–5) and high Mg content, emerges as a promising configuration. This combination seems to effectively reduce the corrosion resistance of the Zn-Mg alloy, enhancing its dissolution and overall electrochemical activity.

The presented observations shed light on the intricate relationships between various parameters and the electrochemical behavior of the alloy specimens. The correlation between the Mg content and electrochemical properties becomes apparent through the observation of potential–time curves of the hybrids. The increase in potential over time for the 3Mg and 10Mg specimens signifies the formation of passive films on their surfaces, while the declining potential for the 30Mg specimen points to higher surface reactivity. This phenomenon aligns with the observed trend of corrosion potential reduction and increased corrosion current density as the Mg content rises. The higher Mg content promotes greater dissolution, leading to more aggressive electrochemical activity. The distinction in nobility between Mg and Zn is a crucial factor that significantly impacts the electrochemical behavior of these alloy specimens. This concept of nobility refers to a metal’s tendency to resist corrosion or oxidation, with nobler metals being less prone to corroding when in contact with a less noble metal. In this case, Mg is considered inherently less noble than Zn, which positions it as the more susceptible candidate to undergo corrosion in a galvanic couple. This fundamental principle serves as the cornerstone for understanding the observed trends. As the alloy becomes richer in magnesium, the proportion of Mg available for corrosion also rises. This increase in available Mg atoms amplifies the potential for electrochemical reactions, resulting in a more pronounced electrochemical response. This explains the observed trend of a decrease in corrosion potential and an increase in corrosion current density as the Mg content increases [60].

The effects of the number of turns on the corrosion properties offer a deeper understanding. The influence of the number of turns on the corrosion properties within the hybrid process indicates that the microstructure’s complexity plays a pivotal role. An increased number of turns enhances the open circuit potential, metastable pitting behavior, and charge transfer resistance, while also enlarging the resistive–capacitive properties in the Nyquist plot (Figure 7 and Figure 8). This is likely a consequence of the interactions between microstructural features and electrochemical phenomena, encompassing metallic dissolution, passivation, deposition of corrosion products, and the subsequent surface coverage by these products. In contrast, for the casting specimens, the number of turns appears to exert minimal influence on potential, corrosion current density, passive domain, charge transfer resistance, and passive film properties, where this can be attributed to the homogeneity maintained in the microstructure due to the uniform distribution of turns. While for the hybrids, reducing the number of turns ensures a distinct separation between the two metals, Zn and Mg. This separation establishes well-defined boundaries and zones between them, enabling better control of their electrochemical and chemical reactions (metallic dissolution and passivation), as the micro-galvanic effects are influenced by the surface area and the arrangement of the anode sites (Mg-rich phases). Additionally, it reduces the concentration of intermetallic compounds [47,49].

Comparisons between the casting and hybrid processes offer valuable insights into their unique corrosion behaviors. The casting specimens exhibit a notably higher degree of corrosion resistance, as evidenced by their low corrosion current densities and high corrosion potential (Table 5). In contrast, the hybrid process yields contrasting outcomes, characterized by higher corrosion current densities and lower corrosion potential. These differences can likely be attributed to the microstructural attributes intrinsic to the hybrid specimens. Specifically, these hybrid specimens feature four chemically distinct phases: Zn, Mg, Mg_2_Zn_11_, and MgZn_2_. It is noteworthy that the corrosion resistance is found to increase in correspondence with the number of turns within the hybrid process. This could be attributed to the role of these intermetallic phases, specifically MgZn_2_ and Mg_2_Zn_11_. As the number of turns increases, so does the concentration of these intermetallic phases [47,49], resulting in a reduction in the potential difference between Zn and Mg. This effect can be attributed to the fact that intermetallic precipitates tend to hold an intermediate potential between Zn and Mg, where the Mg_2_Zn_11_ phase is known to have the highest corrosion resistance among Zn-Mg phases [61], contributing to the observed alteration in corrosion behavior.

The unique behavior of the Zn-30Mg 15-turns hybrid material, characterized by the lowest corrosion potential and highest corrosion current density, underscores the interplay between the Mg content and the number of turns. The annealed Zn-3Mg alloy, showing the opposite trend, with the highest corrosion potential and lowest corrosion current density, likely benefits from grain structure modifications and the absence of precipitates, impacting its corrosion behavior. The applied PDA heat treatment resulted in a significant enlargement in the grain size, exceeding three times the size observed following HPT. Consequently, this enlargement led to a reduction in the density of grain boundaries within the material, and there were fewer available sites for the precipitation of the Mg_2_Zn_11_ phase. Furthermore, this heat treatment also resulted in a diminished concentration of the Mg_2_Zn_11_ phase within the microstructure, as presented in Figure 3c. Consequently, the result was the attainment of a chemically homogeneous microstructure in the material.

Finally, the study of the electrochemical behavior reveals insightful trends. Higher Mg content correlates with decreased corrosion potential and increased current densities, indicative of high susceptibility to corrosion as the result of a galvanic coupling. Notably, the number of turns exhibits varied effects: an insignificant influence on the potential and corrosion current density in casting specimens, yet a significant impact within the hybrid process. Increasing the number of turns results in a high corrosion potential, reduced corrosive activity, and increased charge transfer resistance. Conversely, in hybrid specimens, the hybrid process separates Mg and Zn, producing large anodic areas and allowing better control over their chemical and electrochemical reactions of metallic dissolution and passivation product deposition. Reducing the number of turns reduces the concentration of intermetallic phases that effectively narrow the potential difference between Zn and Mg. Furthermore, the casting process itself contributes to producing higher corrosion resistance compared to the hybrid approach, and the application of heat treatment leads to an enlargement in the grain size, thereby reducing the number of grain boundaries or the available sites for the precipitation of the Mg_2_Zn_11_ phase. Altogether, these findings provide a comprehensive perspective on how the Mg content, number of turns, and processing methods intricately influence the electrochemical characteristics.

## 5. Conclusions

Zn-(3–30 wt.%)Mg alloy and its hybrids were HPT-processed for up to 30 turns. The microstructures after HPT and subsequent PDA were characterized through SEM imaging and EDS analysis. The corrosion behavior was studied as a function of the number of turns for the HPT-processed hybrid and alloy samples, as a function of the starting materials for the alloy and hybrid HPT-processed samples, and as a function of composition for the HPT-processed hybrid, ranging from 3 to 30 wt.%Mg. The results indicated that a greater number of turns results in greater corrosion resistance because of the formation of intermetallic phases. In addition, the hybrid was more corrosion resistant because it tended to form the intermetallics more readily than the alloy due to the inhomogeneous conditions of the materials before the HPT processing as well as the non-equilibrium conditions imposed by the HPT processing. The hybrids with greater Mg contents were less corrosion-resistant due to the addition of Mg leading to less noble behavior and more reactivity.

## Figures and Tables

**Figure 1 materials-17-00270-f001:**
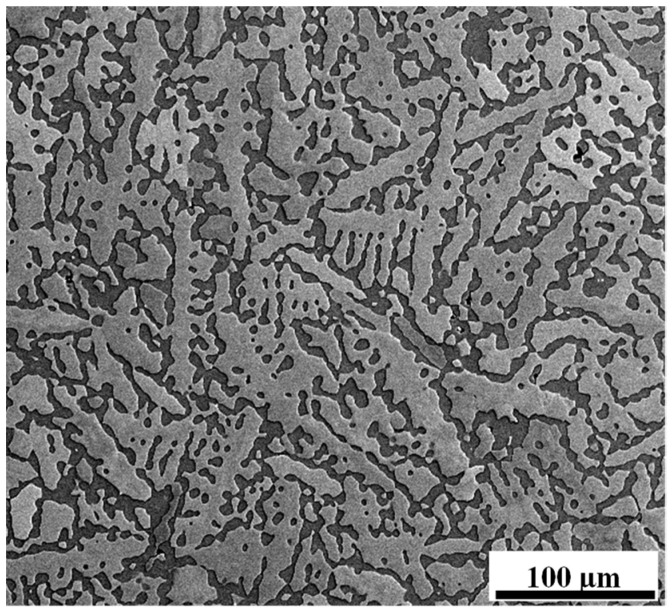
SE-SEM photomicrograph of the Zn-3Mg heat-treated alloy. The darker contrast phase is the Mg-rich phase, and the lighter contrast phase is the Zn-rich phase.

**Figure 2 materials-17-00270-f002:**
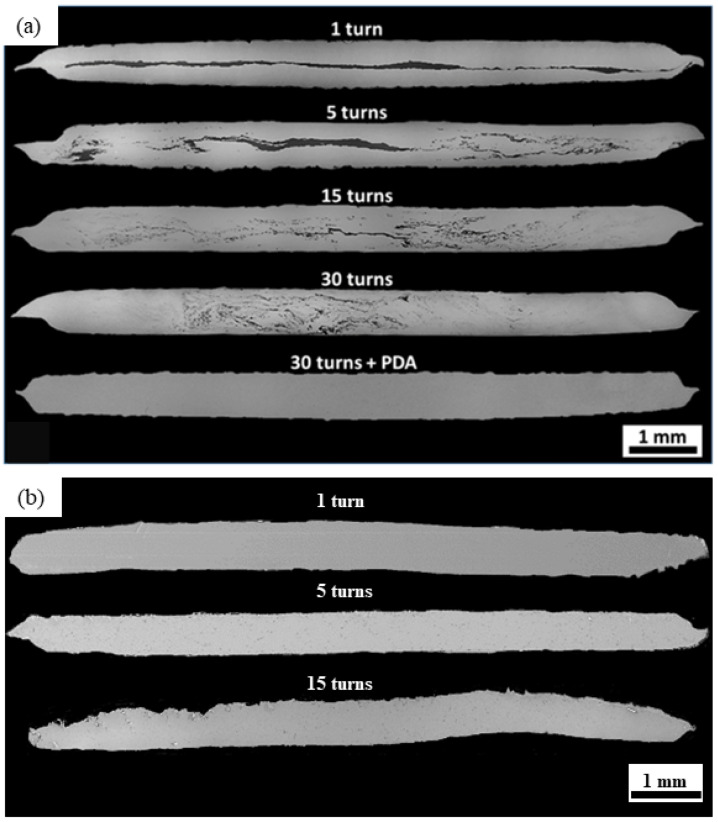
SE-SEM images corresponding to the cross-sections of the HPT Zn-3Mg (**a**) hybrid and (**b**) alloy where the number of turns is indicated and (**a**) shows the 30-turn sample along with the 30-turn sample after post-deformation annealing (PDA). It is noted that the PDA treatment homogenized the distribution of the Mg (dark content) throughout the microstructure compared to the 30-turn sample without the PDA. Adapted from [49].

**Figure 3 materials-17-00270-f003:**
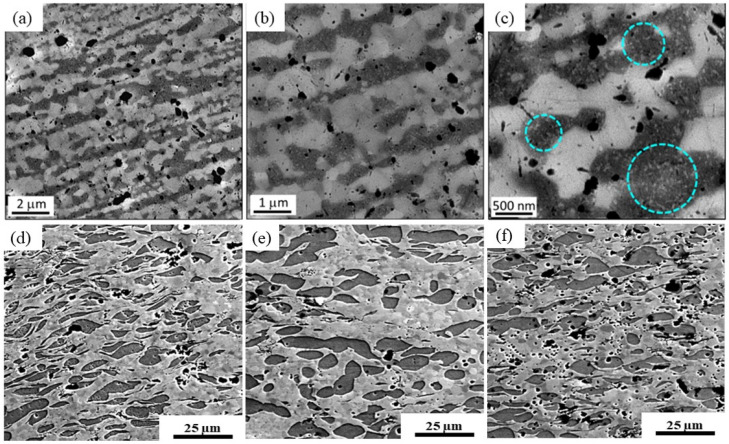
(**a**–**c**) SE-SEM photomicrographs of the Zn-3Mg HPT hybrid near the sample edge after 30 turns of HPT + PDA (200 °C, 1 h). The blue-dashed circles show Mg-rich regions, constituting the Mg_2_Zn_11_ phase. (**d**–**f**) BSE-SEM photomicrographs of the HPT Zn-3Mg alloy near the edge of the sample after 1 turn, at 0.9 mm from the center after 5 turns, and at 1 mm from the center after 15 turns, respectively. The Zn-rich (lighter contrast) and Mg_2_Zn_11_ (darker contrast) phases constituted approximately 68% and 32%, respectively, by volume. Adapted from [49].

**Figure 4 materials-17-00270-f004:**
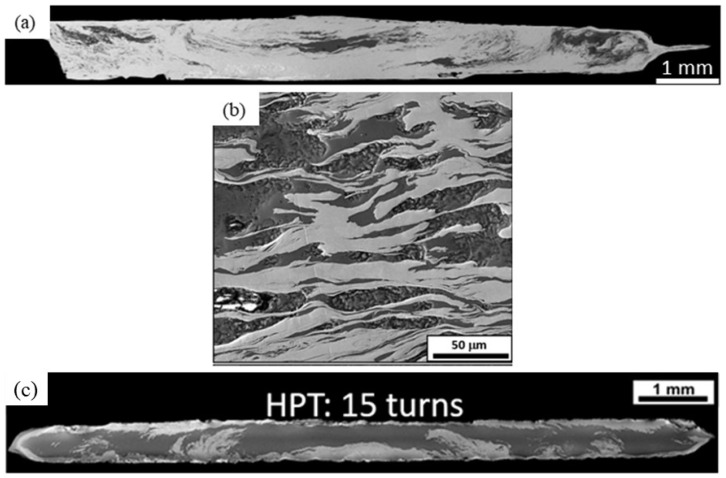
SE-SEM micrograph corresponding to (**a**) low-magnification and (**b**) higher-magnification SE-SEM photomicrographs of the Zn-10Mg HPT hybrid after 15 turns, where image b was acquired near the edge of the disk; (**c**) the Zn-30Mg hybrid after 15 turns.

**Figure 5 materials-17-00270-f005:**
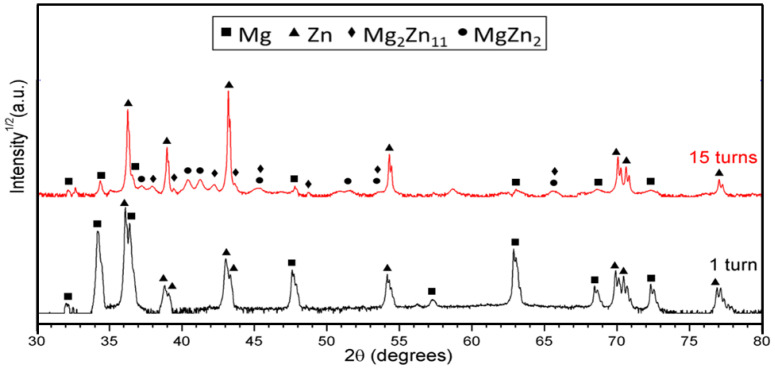
XRD patterns taken from the mid-thickness plane at the periphery of the disk surface area of the Zn-10Mg HPT hybrids after 1 and 15 turns.

**Figure 6 materials-17-00270-f006:**
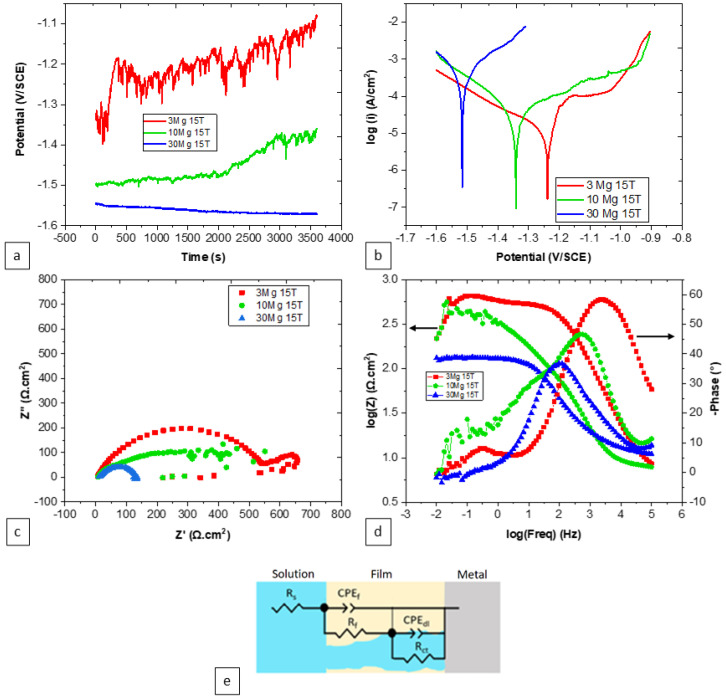
Effect of Mg content on the corrosion behavior of Zn-xMg, where x = 3, 10, or 30, HPT hybrid specimens after 15 turns: (**a**) OCP, (**b**) PDP, (**c**) Nyquist plot, (**d**) Bode plot, and (**e**) equivalent circuit.

**Figure 7 materials-17-00270-f007:**
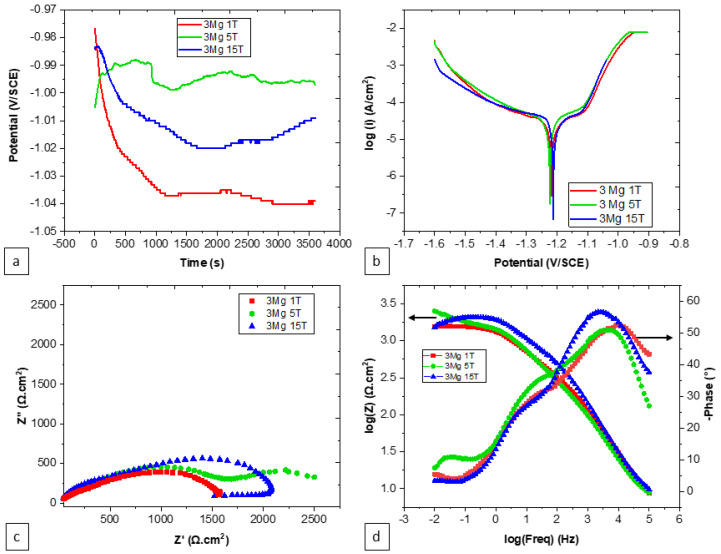
Effect of the number of turns on the corrosion behavior of the Zn-3Mg HPT alloy specimens: (**a**) OCP, (**b**) PDP, (**c**) Nyquist plot, and (**d**) Bode plot. 1T, 3T, and 15T indicate 1 turn, 3 turns, and 15 turns, respectively.

**Figure 8 materials-17-00270-f008:**
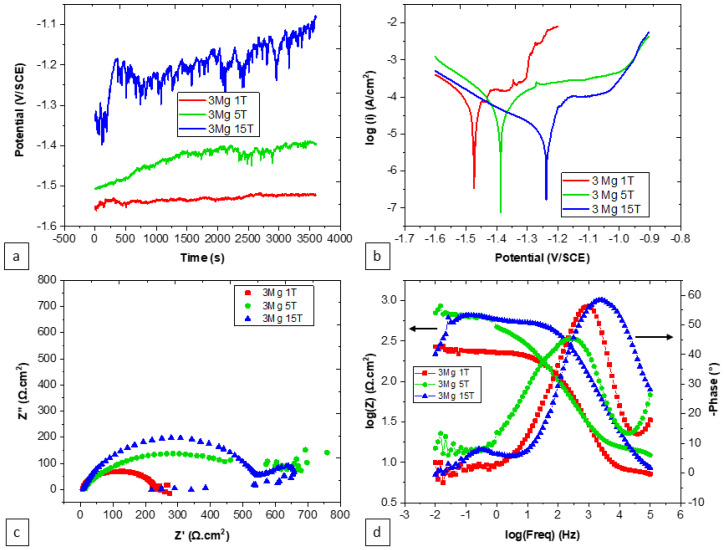
Effect of the number of turns on the corrosion behavior of Zn-3Mg HPT hybrid specimens: (**a**) OCP, (**b**) PDP, (**c**) Nyquist plot, and (**d**) Bode plot. 1T, 3T, and 15T indicate 1 turn, 3 turns, and 15 turns, respectively.

**Figure 9 materials-17-00270-f009:**
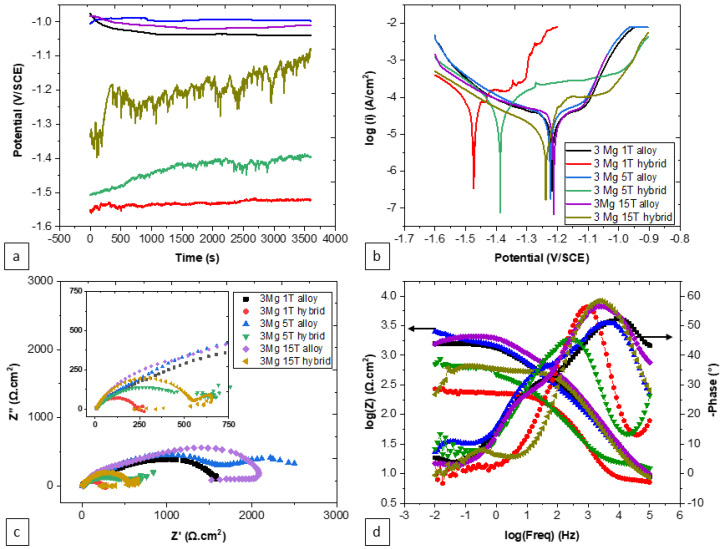
Effect of material processing (HPT alloy versus HPT hybrid) on the corrosion behavior of Zn-3Mg specimens: (**a**) OCP, (**b**) PDP, (**c**) Nyquist plot, and (**d**) Bode plot.

**Table 1 materials-17-00270-t001:** Quantitative results of the Rietveld analysis at the periphery of the Zn-10Mg HPT hybrids after 1 and 15 turns.

Number of Turns	Zn	Mg	Mg_2_Zn_11_	MgZn_2_
	(wt. %)	(at. %)	(wt. %)	(at. %)
1 turn	18.6 ± 0.5	81.4 ± 0.5	0	0
15 turns	57.6 ± 0.4	28.0 ± 0.3	1.6 ± 0.2	12.8 ± 0.4

**Table 2 materials-17-00270-t002:** Electrochemical parameters for HPT hybrids after 15 turns extracted from Figure 6.

Composi-tion	E (V vs. SCE)	i (μA·cm^−2^)	R_s_ (Ω·cm^2^)	Q_f_ (F·cm^−2^·s^(n−1)^)	n_f_	R_f_ (Ω·cm^2^)	Q_dl_ (F·cm^−2^·s^(n−1)^)	n_dl_	R_ct_ (Ω·cm^2^)
Zn-3Mg	−1.19 ± 48 × 10^−3^	13.4 ± 5.5	6.89	80.57 × 10^−6^	0.18	3.9	2.94 × 10^−6^	1	936
Zn-10Mg	−1.35 ± 8 × 10^−3^	40 ± 3.2	7.8	8.08 × 10^−6^	0.77	116	71.02 × 10^−6^	0.62	314
Zn-30Mg	−1.51 ± 4 × 10^−3^	358.3 ± 30.2	5.59	0.24 × 10^−6^	0.8	5.46	4.32 × 10^−6^	0.76	127

**Table 3 materials-17-00270-t003:** Electrochemical parameters of the Zn-3Mg HPT alloy extracted from Figure 7.

Number of Turns	E (V vs. SCE)	i (μA·cm^−2^)	R_s_ (Ω·cm^2^)	Q_f_ (F·cm^−2^·s^(n−1)^)	n_f_	R_f_ (Ω·cm^2^)	Q_dl_ (F·cm^−2^·s^(n−1)^)	n_dl_	R_ct_ (Ω·cm^2^)
1	−1.21 ± 6 × 10^−3^	20.4 ± 2.6	10.66	8.38 × 10^−9^	0.9	391	44.12 × 10^−6^	0.6	1300
5	−1.20 ± 4 × 10^−3^	17.6 ± 4.0	7.8	90.17 × 10^−9^	0.7	1430	16.18 × 10^−6^	0.55	1170
15	−1.19 ± 4 × 10^−3^	24.7 ± 9.3	6.5	4.08 × 10^−9^	0.97	325	80.5 × 10^−9^	0.83	1638

**Table 4 materials-17-00270-t004:** Electrochemical parameters of the Zn-3Mg HPT hybrid extracted from Figure 8.

Number of Turns	E (V vs. SCE)	i (μA·cm^−2^)	R_s_ (Ω·cm^2^)	Q_f_ (F·cm^−2^·s^(n−1)^)	n_f_	R_f_ (Ω·cm^2^)	Q_dl_ (F·cm^−2^·s^(n−1)^)	n_dl_	R_ct_ (Ω·cm^2^)
1	−1.48 ± 6 × 10^−3^	94.6 ± 16.4	6.63	8 × 10^−6^	0.73	3.25	55.4 × 10^−9^	1	213
5	−1.37 ± 11 × 10^−3^	34.6 ± 2.9	6	26.17 × 10^−9^	0.9	14.36	14.58 × 10^−6^	0.69	457
15	−1.19 ± 48 × 10^−3^	13.4 ± 5.5	6.89	80.57 × 10^−6^	0.18	3.9	2.94 × 10^−6^	1	936

**Table 5 materials-17-00270-t005:** Electrochemical parameters of the Zn-3Mg HPT alloy and hybrid extracted from Figure 9.

Material, Number of Turns	E (V vs. SCE)	i (μA·cm^−2^)	R_s_ (Ω·cm^2^)	Q_f_ (F·cm^−2^·s^(n−1)^)	n_f_	R_f_ (Ω·cm^2^)	Q_dl_ (F·cm^−2^·s^(n−1)^)	n_dl_	R_ct_ (Ω·cm^2^)
Alloy, 1	−1.21 ± 6 × 10^−3^	20.4 ± 2.6	10.66	8.38 × 10^−9^	0.9	391	44.12 × 10^−6^	0.6	1300
Hybrid, 1	−1.48 ± 6 × 10^−3^	94.6 ± 16.4	6.63	8 × 10^−6^	0.73	3.25	55.4 × 10^−9^	1	213
Alloy, 5	−1.20 ± 4 × 10^−3^	17.6 ± 4.0	7.8	90.17 × 10^−9^	0.7	1430	16.18 × 10^−6^	0.55	1170
Hybrid, 5	−1.37 ± 11 × 10^−3^	34.6 ± 2.9	6	26.17 × 10^−9^	0.9	14.36	14.58 × 10^−6^	0.69	457
Alloy, 15	−1.19 ± 4 × 10^−3^	24.7 ± 9.3	6.5	4.08 × 10^−9^	0.97	325	80.5 × 10^−9^	0.83	1638
Hybrid, 15	−1.19 ± 48 × 10^−3^	13.4 ± 5.5	6.89	80.57 × 10^−6^	0.18	3.9	2.94 × 10^−6^	1	936

## Data Availability

The data that support the findings of this study are available on request from the corresponding author.

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
