# Peer review of "The Microstructural Evolution and Corrosion Behavior of Zn-Mg Alloys and Hybrids Processed Using High-Pressure Torsion"

_materials, 2024, doi:10.3390/ma17010270_

Round 1

Reviewer 1 Report

Comments and Suggestions for Authors

This article compared the corrosion behavior of Zn-Mg alloys and hybrids with different HPT processing. A series of samples were made, processed, and tested. The experiments were designed properly, the results were valuable. I suggest to publish this article after minor revisions: 

1. In the abstract, add brief background/motivation of this research at the beginning. 

2. Why the sample thickness of HPT alloy and HPT hybrid samples were different? Is there any reason? 

3. The authors described the corrosion test environment. As the authors claimed that these materials were going to use for bio implants or similar situations, what is the corresponding environment in practice?

4. How the torsion tests were conducted? It is not clear in current manuscript. Please describe the equipment, clamps, rotational speed, etc.

5. In fig. 9a and d, the legends were missing.

Author Response

see the attached document

Reviewer 2 Report

Comments and Suggestions for Authors

This study compares the corrosion resistance of a Zn-3Mg alloy and its hybrid counterpart after undergoing high-pressure torsion (HPT) and, alternatively, after subjecting them to HPT followed by post-deformation annealing.

Minimize the use of HPT  in Abstract.

Introduction is more focused on Zn. Why? Add Mg details too.

Figure 1 should be labeled.

What is the criterion for the Zn-rich and Mg-rich?

The study talks about the interactions as a generic term. More details and specifics with examples are required.

Conclusion is weak and not direct. Especially, the first one should be rewritten.

The study is purely experimental. The authors at least can talk about its future work  more than experimentation. What else can be done in analytical and simulation tools?

Author Response

see attached document

Reviewer 3 Report

Comments and Suggestions for Authors

The manuscript entitled "The Microstructural Evolution and Corrosion Behavior of Zn-Mg Alloys and Hybrids Processed Using High-Pressure Torsion" demonstrates Zn-3Mg hybrid generally process better corrosion resistance then Zn-3Mg allogy. Comprehensive materials characterization techniques are used, i.e., SEM, EDX, XRD. The study sheds light in understanding how HPD process aids the grain refinement and phase transformation.

1. Although HPD (high pressure torsion) process results the improvement in grain and hardness, it normally leads to fracture after several turn? Did the authors consider it?

2. Can the authors elucidate more on why after 30 HTT turns, Zn-3Mg alloy gets a homogenous distribution?

3.Figure. 5 needs to include the peaks for ZnO.

4. References are too old, please include recent studies on Zn-3Mg alloy/hybrids.

Comments on the Quality of English Language

English is fine, and no spelling error is detected.

Author Response

see attached document

Reviewer 4 Report

Comments and Suggestions for Authors

I enjoy reading the paper and only have three comments:

1. The authors should check the unit for the scale bar in Figure 2 and Figure 4. I don’t think SEM images could reach such a large scale. If some special methods were used, please indicate them in the manuscript.

2. At line 228, the authors stated that “After PDA, the microstructure showed no macroscopic evidence of different phases.” I don’t SE SEM images could indicate phase information. The authors should show backscattered images or EDS mapping.

3. At line 243, “aftr” should be “after”.

Comments on the Quality of English Language

English is good.

Author Response

see atached document
